# Bisphosphonate nanoclay edge-site interactions facilitate hydrogel self-assembly and sustained growth factor localization

Yang-Hee Kim[1], Xia Yang[2], Liyang Shi[2,3], Stuart A. Lanham[1], Jons Hilborn[2], Richard O.C. Oreffo [1], Dmitri Ossipov [4,5✉] & Jonathan I. Dawson [1,5✉]

Nanoclays have generated interest in biomaterial design for their ability to enhance the mechanics of polymeric materials and impart biological function. As well as their utility as physical cross-linkers, clays have been explored for sustained localization of biomolecules to promote in vivo tissue regeneration. To date, both biomolecule-clay and polymer-clay nanocomposite strategies have utilised the negatively charged clay particle surface. As such, biomolecule-clay and polymer-clay interactions are set in competition, potentially limiting the functional enhancements achieved. Here, we apply specific bisphosphonate interactions with the positively charged clay particle edge to develop self-assembling hydrogels and functionalized clay nanoparticles with preserved surface exchange capacity. Low concentrations of nanoclay are applied to cross-link hyaluronic acid polymers derivatised with a pendant bisphosphonate to generate hydrogels with enhanced mechanical properties and preserved protein binding able to sustain, for over six weeks in vivo, the localized activity of the clinically licensed growth factor BMP-2.

[1] Bone and Joint Research Group, Centre for Human Development, Stem Cells and Regeneration, Institute of Developmental Sciences, University of Southampton, Southampton SO16 6YD, UK. [2] Department of Chemistry, Ångström Laboratory, Polymer Chemistry, Uppsala University, 751 21 Uppsala, Sweden. [3] College of Biology, Hunan University, Changsha 410082, China. [4] Department of Biosciences and Nutrition (BioNut), H2, Karolinska Institute 141 83, Huddinge, Sweden. [5] These authors contributed equally: Dmitri Ossipov, Jonathan I. Dawson. ✉email: dmitri.ossipov@ki.se; jid@soton.ac.uk

Tissue engineering and regenerative medicine seeks to harness the developmental potential of stem cells to replace tissue lost or damaged through injury or disease. Biomaterials such as hydrogels serve in this context to provide a 3D template, or scaffold, able to promote new tissue growth through provision of environmental cues that direct stem cell function. Controlled, highly localized biomolecule presentation in particular, is key for coordinating cell migration, proliferation, and differentiation[1,2]. To date, the traditional drug delivery paradigm of slow release has dominated attempts to control the presentation of biomolecules by hydrogel scaffolds. However, slow release strategies are typically poorly suited to achieving the precisely localized signaling mechanisms at play in natural morphogenesis and present challenges for promoting tissue ingress into spaces occupied by the drug-releasing hydrogel.

Nanoclays offer new opportunities for biomaterial design[3,4]. In particular, nanoclay–protein complexes allow localized activity of bioactive molecules within a hydrogel environment permissive for cellular ingress. This approach has been applied to achieve high loading and localized delivery of insulin-like growth factor-1 mimetic protein[5], localization of vascular endothelial growth factor to initiate the formation of new blood vessels at an injury site[6,7], and localization of bone morphogenetic protein (BMP)-2 to achieve ectopic bone formation at the lowest dose recorded in the literature to date[8]. Furthermore, the interaction of nanoclays with polymers is also of core interest in biomaterial design for their utility as physical cross-linkers that combine the dynamic properties associated with physical hydrogels such as self-assembly and self-healing with greatly enhanced mechanical stiffness and toughness[9–11].

Despite the clear utility of clay–polymer and clay–protein interactions for biomaterial design, the ability to exploit both interactions, simultaneously, has proved challenging. For example, where clay–protein interactions have been applied to modify

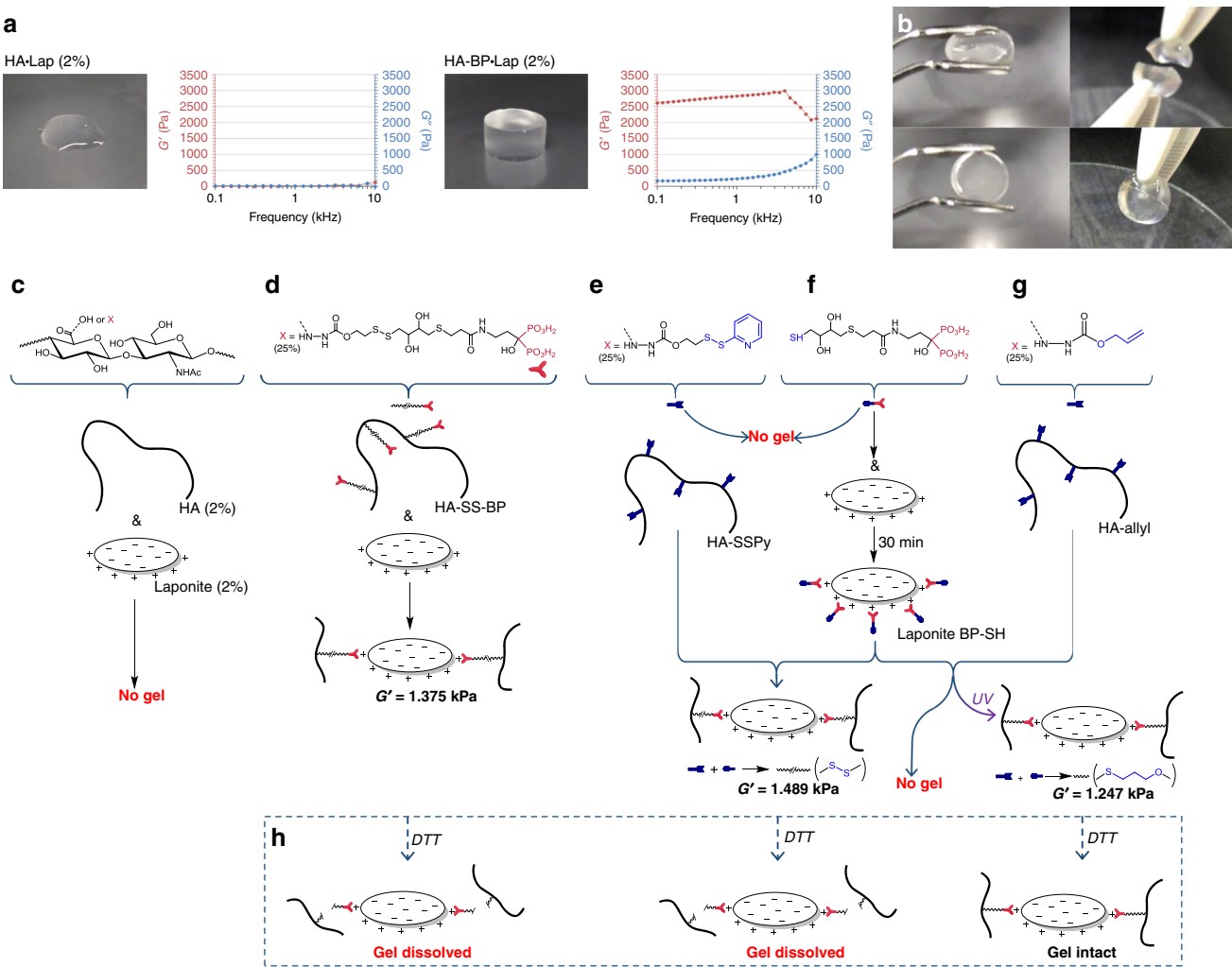

**Fig. 1 Laponite bisphosphonate interactions allow physical cross-linking of hydrogels.** Laponite addition (2 % wt. vol.) to a bisphosphonate (BP) functionalized (DS = 25%) hyaluronic acid (HA) polymer (2 % wt. vol.) yields a stiff gel, while no gel is formed following addition of Laponite to HA alone (**a** graph displays storage ($G'$) and loss ($G''$) moduli of gels under a frequency sweep). HA-BP•Laponite gels can be manipulated using forceps and self-heal (**b**). As well as achieving Laponite cross-linking via BP disulfide functionalization of the HA backbone (HA-SS-BP; **c** and **d**), thiol functionalization of the Laponite particles themselves via incubation with low molecular weight thiol-terminated BP derivatives (BP-SH) was also achieved (**f**). Thiol functionality of Laponite•BP-SH was confirmed via successful gelation of hyaluronic acid modified with dithiopyridyl groups (HA-SSPy) following addition of Laponite•BP-SH, but not of BP-SH derivatives alone (**e**). Thiol functionality of Laponite•BP-SH was also confirmed via successful photoinitiated gelation of allyl-functionalized HA in the presence of Laponite•BP-SH. No gelation of HA-allyl + Laponite•BP-SH was observed in the absence of UV exposure (**g**). Treatment of the formed gels with dithiothreitol (DTT) resulted in dissolution only of those nanocomposite gels where BP groups were conjugated via a labile disulfide bond (**h**). $G'$ measurements were conducted at a frequency of 1 Hz.

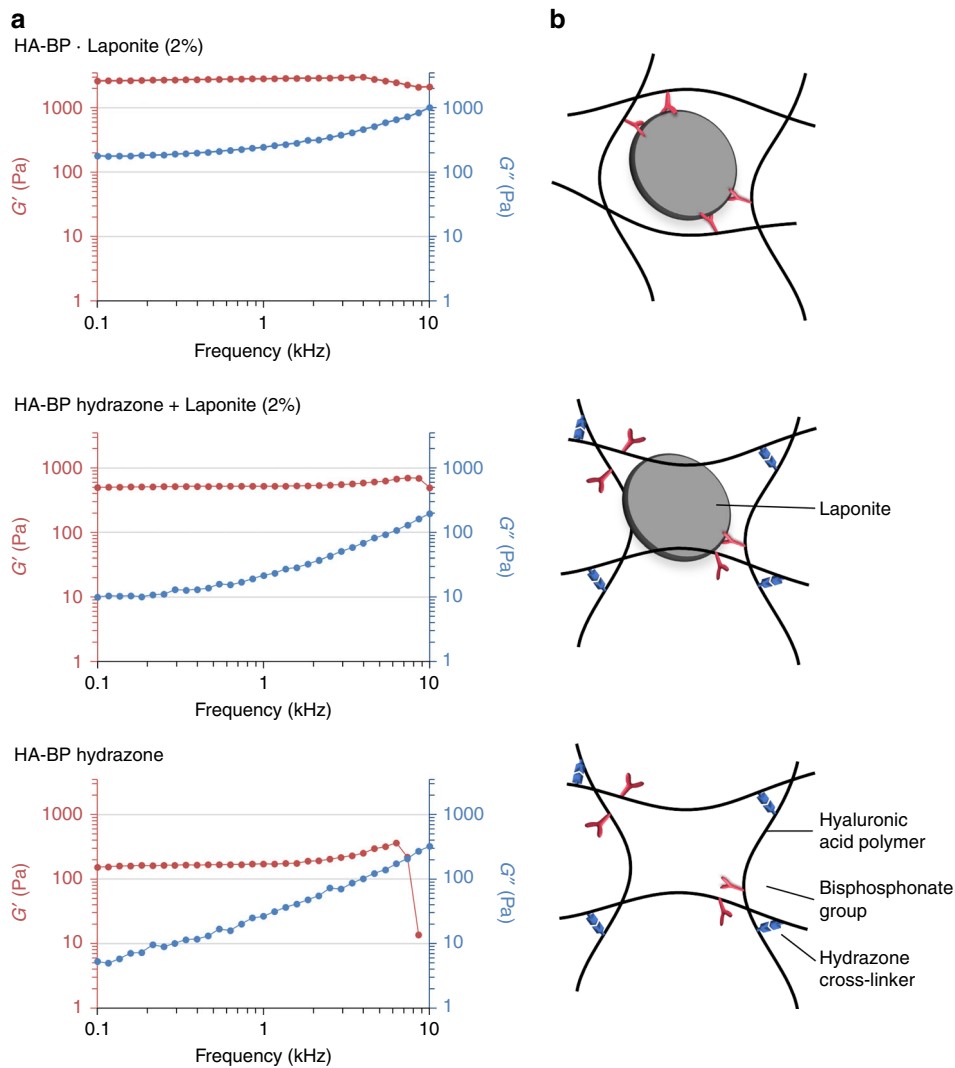

**Fig. 2 HA-BP•Laponite gels are stronger than their chemically cross-linked analogs.** Frequency sweep measurements of storage and loss modulus of HA-BP•Laponite physical hydrogels and chemically (hydrazone) cross-linked HABP analogs without and with clay (**a**). Schematic of proposed interactions (**b**). Interference of the stronger, but slower forming, BP•Laponite cross-links, by weaker, faster forming hydrazone cross-links could account for the weaker gels formed with addition of Laponite to HABP chemical gels compared to HA-BP•Laponite physical hydrogels.

release from a polymeric network, the formation of the clay–biomolecule complex dominates in the design process and minimizes the contribution of the nanoclay phase to the physical properties of the hydrogel[5]. On the other hand, where polymer–clay nanocomposites have been optimized for their mechanical properties, any influence of clay on the drug release profile is often minimal and secondary to the clay's primary influence on the polymeric network (e.g., through reduced swelling or degradation)[9,12]. This competitive compromise between strategies can be explained, at least in part, by the fact that both applications classically seek to employ the same site of interaction: the negatively charged surface of the clay particle. To our knowledge, strategies that utilize the clay edge–site for physical cross-linking of polymers, and the opportunities for material science this approach may afford, have not yet been explored.

LAPONITE$^{TM}$ (Na$_h$(Mg$_{3-h}$Li$_h$)Si$_4$O$_{10}$(OH)$_2$·$n$H$_2$O), a synthetic smectite manufactured by BYK-ALTANA (herein, Laponite) consists of 25–30 nm diameter particles of 1 nm thickness, which possess a permanent negative surface charge and positive (amphoteric) charge in the form of Mg–OH (and some Si–OH) groups on the particle edge[4]. Due to Laponite's small particle diameter compared with natural smectites, the chemistry of the

edge surface (representing ~7.4% of Laponite surface area versus ~0.2% of Montmorillonite) plays a particularly important role in determining the physical properties of Laponite colloids. For example, in the presence of ionic solutes, Laponite particles associate via edge–face interactions to generate gels, or, at high ionic strengths, flocculates. Tetravalent pyrophosphate salts are applied commercially to inhibit aggregation and improve dispersion by complexing with, and thus screening, the hydroxyl groups exposed at the particle edge[13–15].

Bisphosphonate (BP) is the organic analog of pyrophosphate. The additional valency of the carbon atom, which substitutes for the phosphate-bridging oxygen, provides possibilities for conjugation of BPs to functionalize polymers or other low molecular weight molecules[16]. This approach has been applied, for example, to utilize the strong affinity of BPs for bone-bound calcium, facilitating targeting of BP- conjugated drugs or nanoparticles to bone surfaces[17]. Ossipov et al. have extended this concept to generate in situ forming nanocomposite hydrogels by utilizing the coordination of tethered BP ligands with CaP nanoparticles to cross-link a hyaluronic acid (HA) polymer backbone[18,19]. A similar self-assembly approach, applying the affinity of BP for magnesium, has recently been explored using assemblies of free

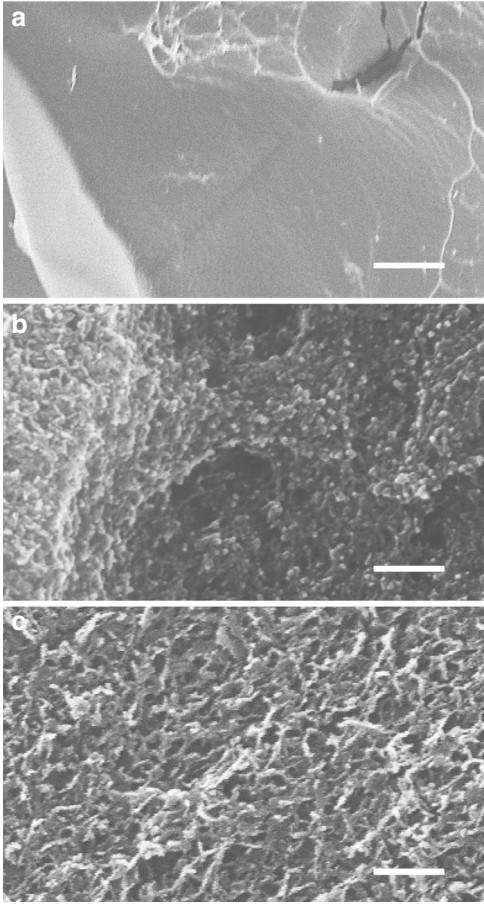

**Fig. 3 Scanning electron microscope characterization of chemical and physical gels.** HABP chemical gels (**a**) compared with HA-(BP)$_3$•Laponite physical gels (**b**) and chemical gels assembled through mixing of pyridyldithio-modified hyaluronan (HA-SSPy) with thiol-functionalized Laponite (Laponite•BP-SH) (**c**). Scale bar = 200 nm.

$Mg^{2+}$ coordinated by small molecular weight BP ligands and HA-bound BP functional groups[20].

Inspired by the well-characterized pyrophosphate–edge interaction, we hypothesized that clay complexation with HA-tethered BP groups could be harnessed to generate self-assembling nanocomposite hydrogels with preserved clay surface reactivity for additional functionality. Thus, we have explored the potential to apply clay–BP interactions to achieve the physical gelation of HA-BP polymers and enhance protein binding within HABP chemical gels.

## Results

**Bisphosphonate interactions with nanoclay edge-sites**. Low weight percentage solutions (1–2%) of Laponite did not form a gel when mixed with the negatively charged HA backbone. Following BP functionalization of HA however, addition of Laponite generated stiff gels displaying elastic and self-healing properties (Fig. 1a–c, Supplementary Fig. 1). Notably, the HA-BP•Laponite physical hydrogels displayed a 15× higher elastic modulus than their chemically (hydrazone) cross-linked HABP analogs in the absence of clay (Fig. 2a). Laponite addition to chemically cross-linked HABP gels led to only a 3-fold increase in modulus (Fig. 2a) indicating interference of the slow forming BP-Laponite coordination (occurring over several hours) by chemical cross-linking and confirming the enhancements conferred by utilizing nanoclay as a cross-linker (Fig. 2b).

Gel storage modulus was dependent on nanoclay concentration and on the nature of the BP attachment chemistry (Supplementary Fig. 2). For example, a gel containing 2% Laponite displayed approximately double the storage modulus of a gel containing 1% Laponite (Supplementary Fig. 2a). Thiol-ene photochemical addition of BP-acrylamide to HA-thiol allowed attachment of approximately three BP groups to one thiol group of HA to produce a brush-like arrangement of BP groups along the HA backbone (termed HA-(BP)$_3$, degree of substitution (DS) = 15%)[18]. This approach generated a gel with a 5× higher storage modulus following the addition of Laponite than gels generated through a disulfide attachment of BP groups to the HA backbone (HA-SS-BP, DS = 25%). The disulfide attachment, as well as being more labile than the acrylamide-thiol linkage, allowed the tethering of only one BP group per side chain. SEM analysis of physically cross-linked HA-(BP)$_3$•Laponite hydrogels following freeze–drying revealed a nanoporous structure of fibrous aggregates which contrasted with the considerably more uniform appearance of freeze dried HABP chemical gels at the same submicron scale (Fig. 3).

To confirm the involvement of Laponite edge sites in the interaction with BP functional groups, Laponite nanoparticles were preincubated with pyrophosphate ions prior to mixing with HA-(BP)$_3$ (Fig. 4a, b). This reduced the stiffness of the gels by half ($G' = 2277$ Pa at 1 Hz), compared with gels prepared in the absence of pyrophosphate ($G' = 5660$ Pa at 1 Hz). This is consistent with competition for nanoclay Mg–OH edge sites. Gel stiffness partially recovered ($G' = 3638$ Pa) following swelling in PBS, indicating the release of pyrophosphate and the resultant formation of new physical cross-links. Notably, HA-BP•Laponite physical gels prepared in pure water were resistant to swelling (Fig. 4a). Preincubation of HA-(BP)$_3$ with BP-competing $Mg^{2+}$ ions did not result in an equivalent weakening, but rather achieved a further increase in gel stiffness ($G' = 6400$ Pa, Fig. 4c), likely due to the interlayer $Mg^{2+}$ cation bridging of BP-bound clay particles. Accordingly, the osmotic release of interlayer $Mg^{2+}$ upon swelling, caused an overall reduction in gel stiffness to below that observed for the gel prepared in pure water. This result reveals that swelling-resistant BP • Laponite cross-links were, in fact, compromised by addition of $Mg^{2+}$ consistent again with competition for bisphosphonate groups.

**Bisphosphonate functionalization of nanoclay edge-sites**. To explore the application of BP–Laponite interactions for nanoclay functionalization, thiolated low molecular weight BP molecules (BP-SH) were incubated with Laponite with the aim of imparting thiol-functionality to Laponite (Fig. 1c). Addition of Laponite•BP-SH to polymeric HA-SSPy derivative resulted in gel formation through a chemical thiol-disulfide exchange reaction, whereas addition of BP-SH without Laponite did not yield a gel. SEM analysis on the freeze dried gels again revealed a nanoporous structure of fibrous aggregates that is not apparent in chemically crosslinked HABP gels in the absence of clay. Interestingly this approach seemed to yield a more uniform pore distribution compared with the physically cross-linked HA-(BP)$_3$•Laponite hydrogels, perhaps due to the improved miscibility of Laponite•BP-SH particles due to edge site OH screening (Fig. 3). To further prove the principle using a second chemistry, a gel was also formed through UV photo-initiated thiol-ene addition reaction of Laponite•BP-SH to allyl-derivatized HA (see structure in Fig. 1c(v)). Again, gelation did not occur in the absence of UV light, confirming the specificity of the reaction. Finally, treatment of the formed gels with dithiothreitol (DTT) resulted in dissolution of those nanocomposite gels where BP groups were conjugated via a labile disulfide bond, but not gels formed through photo-initiated thiol-ene cross-linking, as the thioester

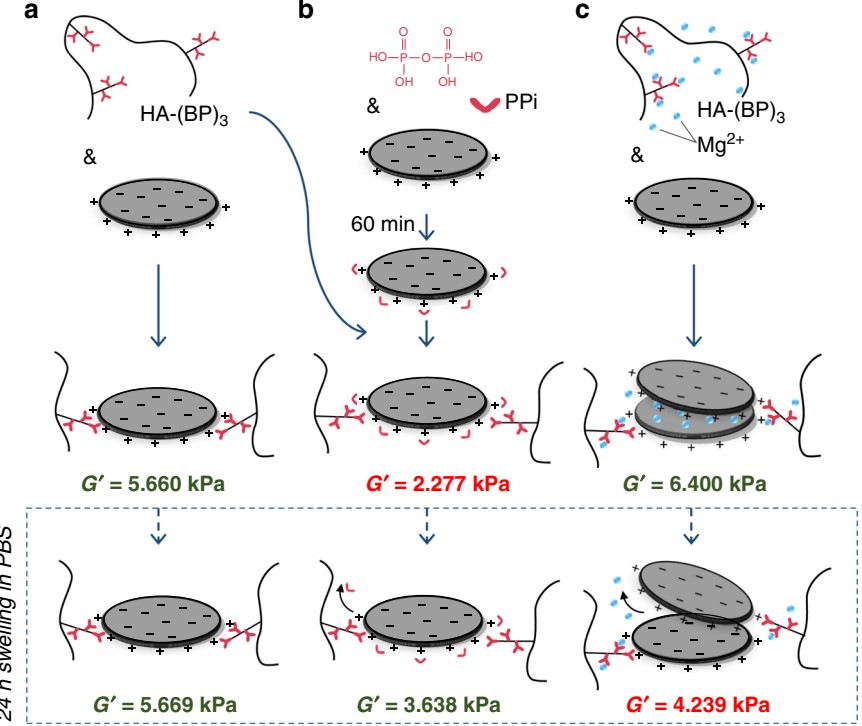

**Fig. 4 HA-BP•Laponite gels are compromised by edge-site or bisphosphonate-associating ions.** Compared with gels prepared in the absence of pyrophosphate (**a**), screening of Laponite edge sites with pyrophosphate ions (PPi) prior to mixing with HA-(BP)$_3$ reduced the stiffness of the gels by half. Stiffness partially recovered post swelling suggesting release of PPi and the formation of new physical cross-links (**b**). Preincubation of HA-(BP)$_3$ with BP-competing Mg$^{2+}$ ions caused an initial increase in gel stiffness consistent with Mg$^{2+}$ cation bridging of Laponite particles. Osmotic release of interlayer Mg$^{2+}$ revealed that swelling-resistant BP•Laponite cross-links (**a**) were compromised by addition of Mg$^{2+}$ (**c**). Storage modulus (G') is recorded at 1 Hz.

bond linking of Laponite•BP to HA is insensitive to DTT. Taken together, these results provide confirmation for the specificity of the clay–bisphosphonate interaction and open up exciting new opportunities for functionalizing Laponite edge sites.

**Protein binding by bisphosphonate-tethered nanoclay.** To explore the potential of BP bound Laponite for growth factor delivery, we examined protein retention and release in vitro and in vivo (Fig. 5). Hydrazone cross-linked HABP gels were prepared with and without Laponite and loaded with either a positively or negatively charged model protein (lysozyme, MW 14.4 kDa; pI 11 and bovine serum albumin, MW 66.5 kDa; pI, 4.7, respectively). Consistent with the preserved availability of Laponite cation exchange sites, no release of loaded lysozyme was detected from HABP + Laponite gels over 7 days. HABP gels alone displayed binding affinity for lysozyme (as in other studies[21]) and the release profile corresponded with HA leaching (in contrast to HABP + Laponite gels from which no release of either Laponite or HA was detected) indicating that an indirect influence of Laponite on hydrogel stability played a role in limiting release. Interestingly, and in contrast to its rapid release from HABP alone, HABP + Laponite gels also restricted the release of nega-tively charged albumin over 7 days. This is consistent with studies that describe patch binding onto Laponite surfaces of negatively charged proteins, including BSA, via adsorption onto clay of local acidic residues exposed on the protein surface[22].

The protein loading capacity of HA-BP•Laponite physical gels was also assessed (Supplementary Fig. 3). Following equilibration in a 10 mg ml$^{-1}$ solution of cytochrome C (Mw 11.7 kDa; PI, 9.6) HA-BP•Laponite gels displayed a very high loading capacity (9.83 mg ml$^{-1}$) compared with HA and HABP hydrazone cross-linked analogs in the absence of clay (1.08 and 3.57 mg ml$^{-1}$,

respectively). Again, negligible release of the loaded protein was observed from the HA-BP•Laponite gels over the subsequent 6 day incubation period (Supplementary Fig. 3b).

To assess the localization of loaded protein in vivo, Cy7-labeled lysozyme was loaded within hydrazone cross-linked HABP gels with and without Laponite (1%) and implanted subcutaneously in mice before scanning of fluorescence over 6 weeks (Fig. 5b). Unbound 1% Laponite solutions were implanted as a further control. Notably, while Cy7-labeled lysozyme was rapidly lost from HABP gels alone, HABP bound Laponite displayed strong and highly localized Cy7 dye intensity even 42 days (6 weeks) after implantation.

To test the ability of Laponite bound protein to promote a regenerative response in vivo, the above study design was repeated, substituting labeled lysozyme for a low concentration (5 µg ml$^{-1}$, 300 ng per gel) of the bone inductive growth factor, BMP-2. Consistent with previous studies, both HABP alone[16] and Laponite alone[8] (here, in contrast to previous studies, delivered as a low concentration sol) were able to achieve detectable bone induction with addition of low dose BMP-2, albeit variably and at low volumes (Fig. 6). In contrast, after 4 weeks, HABP + Laponite gels displayed a synergistic influence on BMP-2 mediated bone induction to achieve significantly ($P < 0.0001$, two-way ANOVA, Dunnett's multiple comparisons test) higher bone volumes compared with all other treatments. Importantly, and unusually for ectopic bone induction at low BMP-2 concentrations[23], significant ($P < 0.0001$, two-way ANOVA, Tukey's multiple comparisons test) increases in bone volume were sustained over the entire 6-week implant period. This result confirmed the retension of active BMP-2 within the nanocomposite hydrogel over time and the ability to achieve robust ectopic bone induction at doses below the typical efficacy threshold for BMP2 delivery strategies reported in the literature (Supplementary Fig. 4). Upon

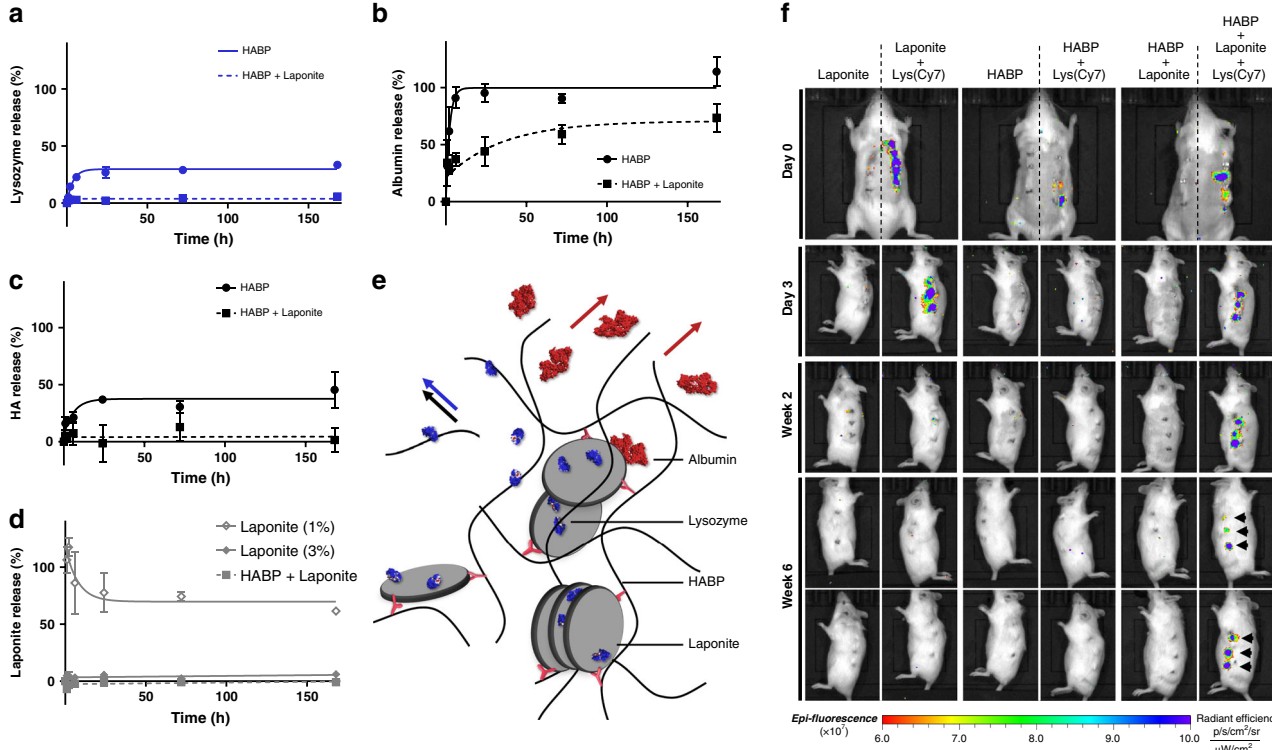

**Fig. 5 Bisphosphonate-tethered Laponite sustains retention of model proteins.** In vitro release profiles of lysozyme (**a**), albumin (**b**), and HA (**c**) from hydrazone cross-linked HABP hydrogels with and without Laponite revealed subdetectable or slowed release with addition of Laponite. Release of Laponite (1%) from HABP hydrogels was compared with 1% and 3% Laponite which present as a solution and gel, respectively (**d**). Hydrogels were incubated in PBS at 37 °C over 7 days incubation. Calculated lower limits of detection for lysozyme, HA and Laponite were 15%, 6.5% and 8.9%, respectively. Error bars = standard deviation (n = 3). **e** Schematic highlights possible modes of Laponite interaction and release of albumin (red arrows), lysozyme (blue arrow) and HA (black arrow)). **f** In vivo release of Cy7-labeled lysozyme from Laponite, HABP, and HABP + Laponite hydrogels after subcutaneous implantation. Negligible Cy7-labeled lysozyme was detected around HABP gel 3 days after implantation, while the lysozyme incorporated into HABP + Laponite gels showed a strong Cy7 dye intensity even 42 days (6 weeks) after implantation.

harvest at 6 weeks, only HABP + Laponite gels remained recoverable from the implant site while HABP in the absence of Laponite had been completely resorbed. Histological analysis revealed that with addition of BMP-2, the hydrogel, largely preserved in the absence of BMP-2, appeared to be almost completely replaced by new bone tissue (Fig. 6). Thus after, 6 weeks in vivo, BP-bound Laponite was able to both stabilize HA hydrogels and bind, protect and sustain localized protein concentrations to enhance bone induction.

## Discussion

In summary, we have demonstrated the potential to exploit bisphosphonate nanoclay edge–site interactions to generate a new class of nanocomposite with enhanced functionality through preserving interlayer or particle surface cation exchange sites (Fig. 7). Using this approach we have generated self-assembling, high water content, negatively charged polymer hydrogels that display dramatic enhancements in mechanical properties. We have also provided proof of concept for the use of low molecular weight bisphosphonated groups to impart edge–site functionality to clay nanoparticles. Furthermore, we have demonstrated the utility of bisphosphonate anchored Laponite to sustain the localization of active proteins in vivo over many weeks to achieve, as proof of concept, bone induction at a greatly reduced effective dose of BMP-2. This new approach to nanocomposite assembly has broad applications across healthcare and beyond.

## Methods

**Preparation of physical hydrogels**. Physical hydrogels were prepared by interaction of HA derivatized with BP groups and Laponite nanoparticles. Two types of HA-BP derivatives were used in which BP groups were linked either through disulfide bonds (HA-SS-BP) or by thiol-ene addition to HA-SH derivative yielding multiple BP ligand attachments (HA-BP$_n$)[24,25]. Full methods for the preparation of HA derivatives are provided in Supplementary Information and Supplementary Fig. 6. Structural characterization of HA-SS-BP derivative was performed by $^1$H-NMR. DS with BP groups was calculated by comparative integration of HA acetamide peak and the ten methylene protons peaks[23]. For fabrication of hydrogels of 0.3 mL by volume, 150 μL 4% solution of HA-BP derivative in water was mixed with 150 μL of freshly prepared solution of Laponite at either 4% or 2% concentration to generate composite hydrogels of 96% and 97% water content, respectively. The mixed solutions were transferred into 2 mL syringes, sealed with Para-film to prevent evaporation and kept overnight for gelation. The formed hydrogels of cylindrical shape (with 8 mm diameter) were removed from the syringe molds. HA concentration in the hydrogels was 2%, whereas Laponite concentration was 2% or 1%. Control experiments were also performed in which HA-BP derivatives were substituted with native HA. The influence of pyrophosphate ions on gelation was studied by preincubation of 8% Laponite solution with equal volume of 0.3 M Na$_4$P$_2$O$_7$ × 10H$_2$O solution for 1 h prior to the mixing with HA-BP derivative. Similarly, for the study of magnesium ions on gel properties, 6% solution of HA-BP derivative (100 μL) was preincubated with 0.4 M solution of MgCl$_2$ (50 μL) prior to the mixing with Laponite solution.

**Preparation of chemically cross-linked hydrogels**. Hydrazone cross-linked HA hydrogels with or without Laponite were prepared by mixing hydrazide and aldehyde-modified HA derivatives. Four types of hydrogels were obtained: (1) HA hydrogel without attached BP groups (HA gel), (2) HA hydrogel without attached BP groups and with encapsulated Laponite NPs (HA + Laponite gel), (3) HA hydrogel with attached BP groups (HABP gel), (4) HA hydrogel with attached BP groups and with encapsulated Laponite (HABP + Laponite gel). Common aldehyde-modified HA derivative (HA-al) was used for all four types of the hydrogels. To obtain the hydrogels from groups 1 and 2, hydrazide-modified HA

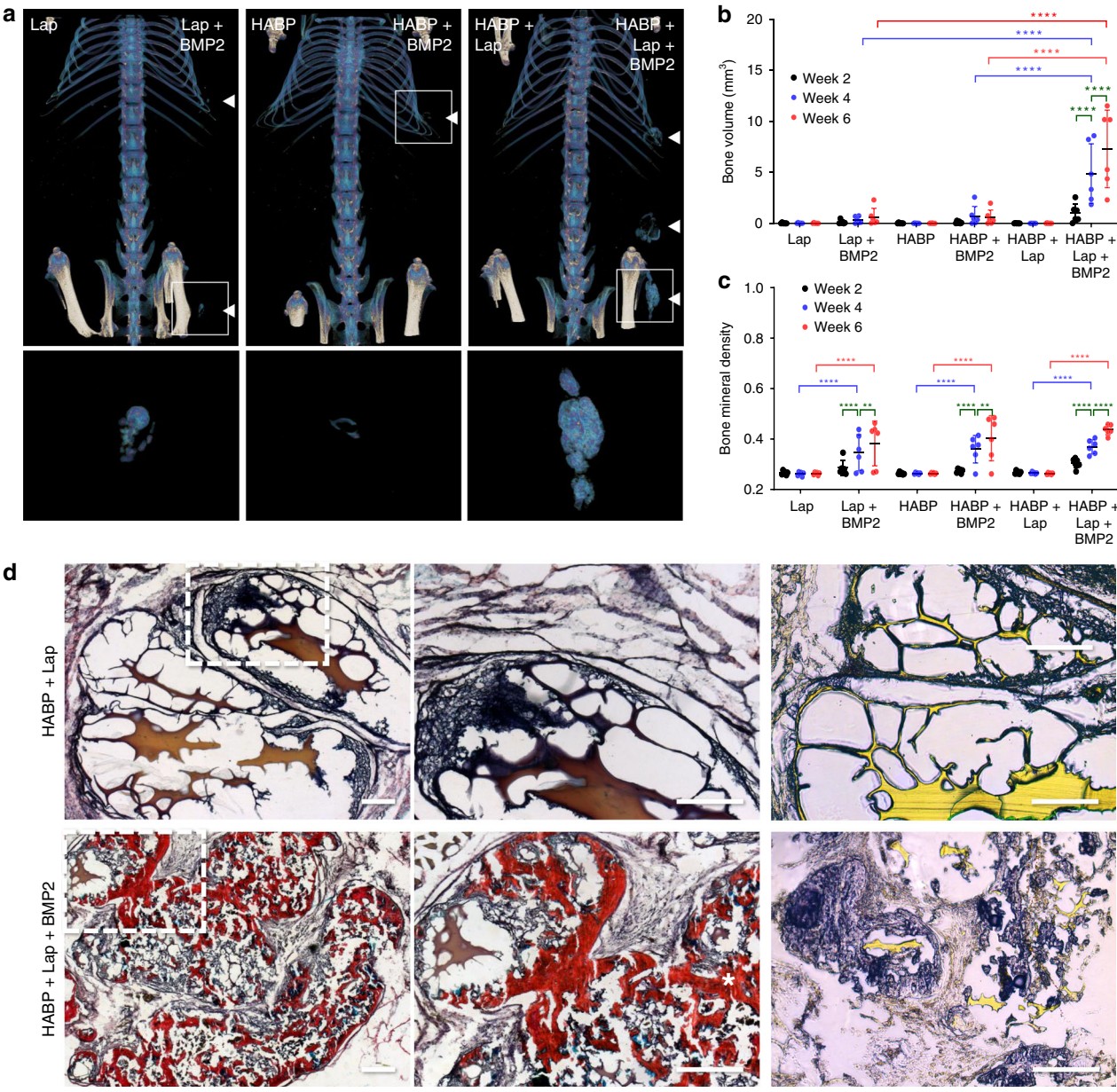

**Fig. 6 Bisphosphonate-tethered Laponite sustains retention of functional BMP-2.** Laponite, HABP, and HABP + Laponite were loaded with low doses (5 μg ml⁻¹) and implanted subcutaneously in mice to assess ectopic bone induction. False color μCT reconstructions of ectopic bone 6 weeks post implantation (**a**). HABP + Laponite delivered BMP-2 to induce significantly higher volumes of bone compared to either HABP or Laponite alone and sustained a significant increase in both bone volume (**b**) and density (**c**) over 6 weeks (error bars = standard deviation, *n* = 6; **, *** and **** indicate *P* values of 0.01, 0.001, and 0.0001, respectively, two-way Anova, Tukey's multiple comparisons test). Histological analysis of the recovered hydrogels revealed extensive regions of Sirius red stained bone tissue only in HABP + Laponite hydrogels incorporating BMP-2 (**d**). Auramine O staining (yellow) of Laponite indicates extensive degradation of the hydrogel structure with addition of BMP-2 (**d**). Scale bar = 50 μm.

derivative (HA-hy) was used, whereas to obtain the hydrogels from groups 3 and 4, HA dually modified with hydrazide and BP groups (BP-HA-hy) was used. Firstly, solutions of Laponite and the HA components in water were prepared at 4% and 3%, respectively. The HA + Laponite and HABP + Laponite hydrogels of 0.3 mL volume were obtained by mixing of 25 μL of water with 75 μL of Laponite, 100 μL of HA-al, and 100 μL of hydrazide component (either HA-hy or BP-HA-hy) subsequently. For preparation, of HA and HABP hydrogels, Laponite was substituted with water. The mixtures were quickly transferred into syringe molds to prevent evaporation and kept overnight, as described for the preparation of physical hydrogels.

**Hydrogel characterization.** Rheological characterizations of all hydrogels were performed using an AR2000 Advanced Rheometer (TA Instruments) with aluminum parallel plate geometry of 8 mm diameter. Frequency sweeps from 0.1 to 10 Hz were performed by monitoring storage (*G′*) and loss moduli (*G″*) at a fixed

normal force (0.015 N) and a fixed strain. Frequency oscillation sweep from 0.1 to 10 Hz was performed by monitoring storage (*G′*) and loss moduli (*G″*) at fixed strain of 1%. Strain oscillation sweep from 0.2 to 100% was performed by monitoring storage (*G′*) and loss moduli (*G″*) at fixed frequency of 0.5 Hz. Time oscillation sweep was performed by applying strain at low (0.2%) and high (50%) values in an alternating manner but at the fixed frequency of 0.5 Hz. All experiments were repeated three times. Rheology plots represent measurements taken from the same sample. Mechanical properties were studied for as prepared hydrogels as well as after their equilibrium swelling in PBS for 24 h. For scanning electron microscopy (SEM) analysis hydrogels were freeze-dried and coated with 8–10 nm of Au/Pd before analysis on SEM (LEO 1550, Zeiss).

**In vitro release of proteins and degradation of hydrogels.** In vitro release of lysozyme (egg white) and albumin (bovine serum) from hydrazone cross-linked HABP hydrogels with or without Laponite and their degradation were evaluated.

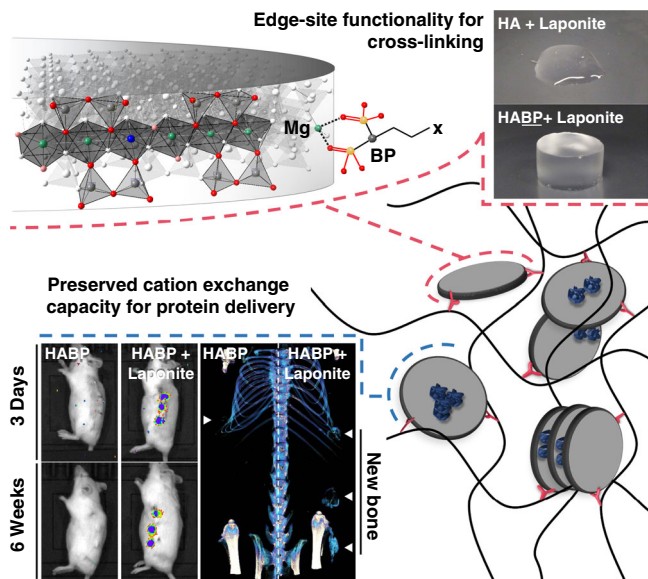

**Fig. 7 Clay–bisphosphonate complexation for dual functionality.**
Clay–bisphosphonate interactions can generate functionalized nanoparticles and self-assembling nanocomposite hydrogels while preserving clay surface reactivity for additional functionality. Clay–bisphosphonate interactions can be harnessed to generate high water content hyaluronic acid hydrogels with greatly enhanced mechanical properties and the capacity to bind the bone inducing growth factor, BMP-2, for enhanced localized efficacy in vivo. Adapted with permission from[9].

BP-HA-hy, HA-al, and Laponite solutions were prepared in deionized water (DW) before mixing to make 2% HABP hydrogel and 1% Laponite as described above. Protein solutions (60 mg ml⁻¹) were added to the HABP ± Laponite pre-gel solution to give a final concentration of 750 μg ml⁻¹ (15 μg protein per 20 μl gel) before incubation. To limit pre-gelation interactions between the protein and HABP or Laponite, all the solutions including protein, were mixed in a single step using gentle vortexing prior to gelation. We note, however, a pilot release study assessing the effect of component mixing order indicated equivalent protein release profiles irrespective of the method used (Supplementary Fig. 5). In all, 1% and 3% Laponite solutions were also prepared as controls for Laponite release. Twenty microlitre of the pre-gel solutions ($n = 4$) were dropped into 1.5 ml eppendorf Lo-bind tubes (Sigma-Aldrich, UK) and incubated at 37 °C for 1 h to allow gelation to occur. One millilitre of PBS was added into the tubes, followed by further incubation at 37 °C for 7 days. During incubation, the supernatants were collected at each time point (1, 2, 6 h and 1, 3, and 7 days) and stored for further analysis. Lysozyme and albumin concentration in the supernatants was measured using a FluoroProfile® Protein Quantification Kit (Sigma-Aldrich, UK) according to the manufacturer's instruction. The degradation of HA based hydrogels was assessed using a uronic acid carbazole assay[26]. Briefly, 50 μl of the supernatants was placed in a 96-well plate and then 200 μl of 25 mM sodium tetraborate in sulfuric acid was added followed by heating the plate at 100 °C for 10 min. After cooling the plate for another 10 min, 50 μl of 0.125% carbazole solution in ethanol was added and the plate was heated again at 100 °C for 10 min before cooling. Absorbance of the solutions in the plate was measured at a wavelength of 550 nm using a microplate reader (GloMax, Promega Co., UK). For Laponite detection, 25 μl of the supernatants and 50 μl of phenolic auramine solution (Sigma-Aldrich, UK) were placed in a 96-well black plate and fluorescence measured at 475 excitation and 550 emission. All the concentrations of protein, HA, and Laponite were determined based on their standard curve in comparison with blanks. The lower limits of detection (LOD) were calculated for the standard curves of lysozyme, HA, and Laponite where release from HABP + Laponite gels was below the detection limits of the assay. LOD was calculated as follows:

$$\text{LOD} = 3.3 s / \sigma, \tag{1}$$

where $s$ is the standard deviation of the intercept of the response (fluorescence) and $\sigma$ is the slope of the curve[27].

**In vivo release of lysozyme from hydrogels.** All animal studies were carried out following approval from the local Animal Welfare and Ethics Review Board (AWERB) University of Southampton and carried out in accordance with the guidelines and regulations stipulated in the Animals (Scientific Procedures) Act,

UK 1986 under the approved Home Office Project license (PPL 30/2880). Female MF1 wild type mice were anaesthetized with an intra peritoneal injection of a mixture of hypnorm and hypnovel (1:1). 18 μl of HABP, Laponite, and HABP + Laponite hydrogels with 2 μl of Cy7 labeled lysozyme (Nanocs Inc., USA) or 2 μl of DW were placed in a 1 ml syringe mold. Totally, 2% HABP, 1% Laponite, and 2% HABP + 1% Laponite hydrogels with or without Cy7 labeled lysozyme were implanted into back subcutis ($n = 6$). Each mouse received three hydrogels on the left side and three hydrogels incorporating Cy7 labeled lysozyme. As a control, Cy7 labeled lysozyme solution was injected. To assess in vivo release of the labeled lysozyme, mice were scanned immediately after injection of the samples (initial lysozyme intensity) using an in vivo imaging system (IVIS, Perkin Elmer, Hopkinton, MA) at 660 nm (excitation) and 710 nm (emission) wavelengths. At each time point (3 days and 1, 2, 4, and 6 weeks), mice were scanned before sacrifice at week 6 and final scanning (lysozyme intensity post implantation).

**In vivo ectopic bone formation.** Totally, 60 μl of 2% HABP, 1% Laponite, and 2% HABP + 1% Laponite hydrogels with or without BMP-2 (5 μg ml⁻¹) were placed in a 1 ml syringe mold before subcutaneous implantation as above ($n = 6$). At week 0, 2, 4, and 6, the mice were scanned using a microcomputed tomography (micro-CT, Skyscan 1176, Bruker, Kontich, Belgium) with 45 kV, 556 μA, 0.2 mm Al filter and a pixel size of 18 μm. Images were reconstructed using NRecon software with correction for misalignment and ring artefacts. To quantify the bone volume of each samples, the reconstructed images were analyzed using CTAn software. CTvox was implemented to create and visualize the 3D models of samples.

The samples were collected after sacrifice 6 weeks post implantation and were fixed in 4% paraformaldehyde in PBS at 4 °C for 3 days. Subsequently, the fixed samples were embedded in paraffin and sectioned (7 μm). Alcian blue, Sirius red, and Auramine O staining was performed according to standard protocols. Images were captured with an inverted light microscope (Zeiss).

**Statistical analysis.** All statistical analysis was performed using GraphPad Prism 7.0. Results are expressed as the mean ± SD and plotted using the same software. One-phase decay curves were fitted to in vitro release data. Comparisons (two-sided) between in vivo experiment groups were performed using a two-way ANOVA and Tukey's multiple comparisons test.

**Reporting summary.** Further information on research design is available in the Nature Research Reporting Summary linked to this article.

## Data availability

The data that support the findings of this study are available from the corresponding author upon reasonable request.

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

## Acknowledgements

This work was supported by Jonathan Dawson's EPSRC fellowship (grant number EP/L010259/1), a European Community Seventh Framework Program Grant, BioDesign (262948) to Hilborn, Ossipov, and Oreffo, and a Regenerative Medicine Platform Acellular/Smart Materials—3D Architecture (MR/R015651/1) to Dawson and Oreffo. Dmitri Ossipov acknowledges financial support from the Swedish Research Council (Grant No. 2017-04651). The authors would like to thank Dr Janos Kanczler and Ms. Julia Wells of the University of Southampton Bone and Joint group for technical support and Mr. Mohamed Mousa and Ms. Roxanna Ramnarine Sanchez for helpful discussions and critical feedback on the paper.

## Author contributions

Y.K. designed and carried out the in vitro and in vivo protein release and bone induction studies with assistance from S.L. for micro-CT analysis. D.O. carried out the hydrogel design, synthesis, and characterization with assistance from X.Y. and L.S. The original conception of the project and funding was through J.D., R.O., and J.H. The conceptual design of the nanocomposite strategy was by J.D and D.O. who, with Y.H. also wrote the paper with the assistance from all other coauthors. All authors have given approval to the final version of the paper.

## Competing interests

The authors declare the following competing interests. The underlying technology has been patented and licensed to a University of Southampton spin-out company. Jonathan Dawson, Jons Hilborn, Dmitri Ossipov, and Richard Oreffo are shareholders in Renovos. The remaining authors declare no competing interests.
