## [Peer Review File · Nature Communications]

Reviewers' Comments:

Reviewer #1:

Remarks to the Author:

This manuscript proposed an approach to generate a class of nanocomposite with enhanced functionality through preserving interlayer or particle surface cation exchange sites by using the interaction between bisphosphonate (BP) and Laponite nanoclay edge-site. Using this method, the self-assembling, negatively charged polymer hydrogels with enhanced mechanical properties and high protein (such as BMP-2) loading capacity can be constructed. Interestingly, the utility of bisphosphonate anchored Laponite can achieve the sustainable localization of active proteins in vivo over many weeks and bone induction at a greatly reduced effective dose of BMP-2. There are some issues that should be addressed. My specific comments are appended below:

1. On Page 4, authors claimed that the self-healing gels can be obtained after adding the Laponite into the bisphosphonate (BP)-functionalized hyaluronic acid (HA), and provided pictures to show the self-healing behavior of the gel (Figure 1b, page 18), however, some forces should be applied to the healed gel to reflect the healing effect. In addition, to further demonstrate the self-healing properties quantitatively, the self-healing efficiency of the formed gels should be calculated.

2. On Page 5, authors claimed that the thiol-ene photo-chemical addition of BP-acrylamide to HA-thiol allowed attachment of approximately three BP groups to one thiol group of HA to produce a brush-like arrangement of BP groups along the HA backbone; however, relevant evidence, such as structural characterization, was not given. Also, the calculation method to obtain degree of substitution of HA-(BP)₃ and HA-SS-BP should be provided.

3. On Page 20, Figure 3d, the histological analysis was employed to demonstrate the formation of new bone tissue and the degradation of the HABP+Laponite gels, however, the scale labels were missing.

4. In Supplementary figure 3, both the HA-(BP)₃+Laponite hydrogel and HA-(BP)₃+PPi+Laponite hydrogel were incubated in PBS for 24 hours. The G' of the former did not change while the latter increased a bit. How was the stability of these hydrogel in PBS for a longer time? Would it be dissolved or not? How was the stability of the nanoclay composite hydrogel in vivo?

5 Page 12, In the part of in vitro release of proteins and degradation of hydrogels. Would the nanoclay composite hydrogel be completely dissolved in PBS at 37°C? How long would this hydrogel be degraded in vivo?

6 In the conclusion, the author writes that the hydrogel has a high water content, but the water content is not given in the article.

Reviewer #2:

Remarks to the Author:

Nanoclays have been used to both cross-link polymers and to facilitate the sustained localization of therapeutic proteins. The authors of this paper note that biomolecule-clay and polymer-clay interactions are often set in competition, potentially limiting the functional enhancements achieved. To address this problem, the authors apply bisphosphonate interactions with the positively charged clay particle edge to develop a new class of self-assembling hydrogels containing functionalised clay nanoparticles with preserved surface exchange capacity. The authors claim this enables the generation of hydrogels with "dramatic enhancements in mechanical properties and protein binding". I would like to see the authors address the following comments:

1. The authors refer to a previous study [ref 5] that uses laponite to deliver BMP-2 to promote ectopic bone formation. Did the authors use the same dose of BMP2 in this study? What

improvements in bone formation, if any, do the authors observe in the present study compared to reference [5]?

2. The stated problem being addressed is that the nanoclay is typically either used to crosslink the polymer, or to control the release of biomolecule, but typically cannot efficiently enable both as the two utilise the negatively charged surface of the clay. While the solution utilised by the authors is elegant, could the same outcome have been achieved by simply crosslinking the HA hydrogel using an alternative mechanism (e.g. chemical crosslinking of pendant reactive groups or by radical polymerization) and integrating laponite into the crosslinked HA hydrogel to control the release of biomolecule? It is not clear what benefits utilising the clay edge-site for physically crosslinking the HA offers over other strategies for crosslinking HA (e.g. improved mechanical properties over chemical crosslinking; more reliable or specific biomolecule release kinetics over simply encapsulating laponite into a chemically crosslinked HA hydrogel).

3. Does utilising the clay edge-site for physically crosslinking the HA influence the capacity of the system to bind and release negatively charged biomolecules?

4. The authors point to the challenges of "promoting tissue ingress into spaces occupied by the drug-releasing hydrogel". Did the authors assess the degradation rate of the biomaterial?

5. Please discuss any limitations of utilizing laponite in clinical contexts.

Reviewer #3:

Remarks to the Author:

The manuscript "Giving nanocomposites the edge: Bisphosphonate nanoclay edge-site interactions facilitate hydrogel selfassembly and sustained growth factor localization" is a good paper and it is focused on the use of low concentrations of Laponite to cross-link hyaluronic acid polymer derivatized with a pendant bisphosphonate resulting in self-assembling hydrogels with a potential to achieve bone induction at log orders below the current clinical dose.

However, the following issues must be addressed by the authors:

1. The loading procedure of the lysozyme into the hydrogels prior to in vitro drug release studies is not clear. There is a need for detailed information.

2. I suggest that more studies should be performed such as SEM analysis so as to study the hydrogel microstructure, FTIR, cytocompatibility studies, calcium deposition on the hydrogel etc.

3. Were the hydrogels allowed to dry after gelation in the syringe mould? What is the nature of the hydrogels?

4. The authors need to compare their results with the results reported by researchers that designed similar hydrogels for bone regeneration.

5. Did the hydrogel average strength (MPa) fall within the range suitable for bone regeneration?

6. I suggest that the title should be revised.

In response to reviewers

Reviewer #1:

1. On Page 4, authors claimed that the self-healing gels can be obtained after adding the Laponite into the bisphosphonate (BP)-functionalized hyaluronic acid (HA), and provided pictures to show the self-healing behavior of the gel (Figure 1b, page 18), however, some forces should be applied to the healed gel to reflect the healing effect. In addition, to further demonstrate the self-healing properties quantitatively, the self-healing efficiency of the formed gels should be calculated.

We thank the reviewer for this helpful recommendation and have now undertaken this characterisation more formally. Oscillatory strain sweep and four-cycle strain tests were performed and variation of the storage (G') and loss (G'') moduli assessed. This analysis **now presented in Supplementary figure 1** further confirms the self-healing ability of the hydrogel formed by bisphosphonate (BP)-functionalized hyaluronic acid (HA) and Laponite nanoparticles. Notably, self-healing efficiency under these conditions averaged 95% (+/-1.2%).

2. On Page 5, authors claimed that the thiol-ene photo-chemical addition of BP-acrylamide to HA-thiol allowed attachment of approximately three BP groups to one thiol group of HA to produce a brush-like arrangement of BP groups along the HA backbone; however, relevant evidence, such as structural characterization, was not given. Also, the calculation method to obtain degree of substitution of HA-(BP)₃ and HA-SS-BP should be provided.

We thank the reviewer for this helpful point. Hyaluronic acid functionalized with bisphosphonate groups through radical thiol-ene addition of thiol-modified HA (HA-SH) to acrylamide derivative of pamidronate (HA-(BP)₃) was first presented in *Biomaterials*, 2014, 35(25), 6918-6929 (reference [18]). The preparation of hyaluronic acid functionalized with bisphosphonate groups through disulfide exchange reaction (HA-SS-BP) is described in *Advanced Functional Materials*, 2017, 27(37), 1700591 (Reference [23]). In the present work we followed the established protocol for structural characterization using elemental analysis and NMR **which we now describe with citations in the methods**.

3. On Page 20, Figure 3d, the histological analysis was employed to demonstrate the formation of new bone tissue and the degradation of the HABP+Laponite gels, however, the scale labels were missing.

We are grateful to the reviewer for bringing this to our attention. The scale bars are now defined in the legend of figure 3.

4. In Supplementary figure 3, both the HA-(BP)₃+Laponite hydrogel and HA-(BP)₃+PPI+Laponite hydrogel were incubated in PBS for 24 hours. The G' of the former did not change while the latter increased a bit. How was the stability of these hydrogel in PBS for a longer time? Would it be dissolved or not? How was the stability of the nanoclay composite hydrogel *in vivo*?

We thank the referee for this comment. Please note, the purpose of the 24 hour PBS incubation described in figure 3 was not to explore degradation of the hydrogel for which we agree a longer time-frame would be required. Rather our purpose was to assess any effect of swelling on the interference of PPI on the Laponite – bisphosphonate interactions underlying the gelation of HA-BP + Laponite. As described in the manuscript: "Gel stiffness partially recovered ($G' = 3638$ Pa) following swelling in PBS, indicating the release of PPI and the resultant formation of new physical cross-links." An analysis of gel degradation in PBS over a longer time frame is provided in figure 2 and the stability of the HA-BP + Laponite gel over 6 weeks *in vivo* is revealed through auromine O staining of recovered implants in figure 3d (see also below).

5 Page 12, In the part of *in vitro* release of proteins and degradation of hydrogels. Would the nanoclay composite hydrogel be completely dissolved in PBS at 37°C? How long would this hydrogel be degraded *in vivo*?

As indicated in figure 2, there is negligible release of either hyaluronic acid or laponite from the composite hydrogel into PBS over 7 days at 37°C. This is in marked contrast to the HABP chemically cross-linked analogue or the equivalent Laponite concentration in the absence of HABP. As described in the manuscript, histological analysis of composite hydrogels after 6 weeks *in vivo* "revealed that with addition of BMP-2, the hydrogel structure, largely preserved in the absence of BMP-2, appeared to be almost completely replaced by new bone tissue"

6 In the conclusion, the author writes that the hydrogel has a high water content, but the water content is not given in the article.

The water content of the hydrogels has now been added into the section: "Preparation of physical hydrogels"

Reviewer #2:

1. The authors refer to a previous study [ref 5] that uses laponite to deliver BMP-2 to promote ectopic bone formation. Did the authors use the same dose of BMP2 in this study? What improvements in bone formation, if any, do the authors observe in the present study compared to reference [5]?

While our purpose was not to compare the efficacy of the composite strategy with our previously described approach (ref [5], Gibbs et al. 2016 now ref [7]) to deliver BMP2 - which relied on a substantially higher concentration of Laponite to generate a gel (as noted in the manuscript on page 8)- encouragingly the results here compare well with our previous data. Notably our approach provides the ability to induce robust ectopic bone induction at doses below the typical efficacy threshold reported in the literature (0.3–5 µg total dose – reviewed in Gibbs et al. 2016). As in our previous work [5] where ectopic bone induction was detectable at doses between 500 ng and 40 ng, here robust ectopic bone formation was achieved at a dose of 300 ng BMP2. This, together with the unparalleled ability to localise BMP2 at the implant site, provides very strong evidence that the use of Laponite edge sites for cross-linking does not impede its utility for growth factor delivery. **To address this comment (and Reviewer #3's comment [4] below) we now include an additional figure setting this result in the broader context of BMP2 dose reduction strategies (Supplementary figure 7) to further highlight the significance of this result.**

2. The stated problem being addressed is that the nanoclay is typically either used to crosslink the polymer, or to control the release of biomolecule, but typically cannot efficiently enable both as the two utilise the negatively charged surface of the clay. While the solution utilised by the authors is elegant, could the same outcome have been achieved by simply crosslinking the HA hydrogel using an alternative mechanism (e.g. chemical crosslinking of pendant reactive groups or by radical polymerization) and integrating laponite into the crosslinked HA hydrogel to control the release of biomolecule? It is not clear what benefits utilising the clay edge-site for physically crosslinking the HA offers over other strategies for crosslinking HA (e.g. improved mechanical properties over chemical crosslinking; more reliable or specific biomolecule release kinetics over simply encapsulating laponite into a chemically crosslinked HA hydrogel).

The reviewer correctly recognises our aim to simultaneously harness nanoclays for their utility as physical cross-linkers and for their utility in growth factor delivery, but queries if encapsulating laponite within a chemically cross-linked hydrogel could achieve the same end. While this approach may confer the ability to retain growth factors within the gel, it would not impart the dramatic mechanical improvements derived from utilising laponite as a physical cross-linker. This is evident from supplementary figure 1 ("*HA-BP•Laponite physical gels are stronger than their chemically cross-linked analogues with and without nanoclay addition*") which shows a five-fold increase in the storage modulus of physically cross-linked composites over those in which Laponite is encapsulated into the chemically cross-linked analogue. **We have adapted the text to draw out more explicitly the significance of this result (p.5).**

3. Does utilising the clay edge-site for physically crosslinking the HA influence the capacity of the system to bind and release negatively charged biomolecules?

This is an important question that we sought to address through studying the release of albumin (pI= 4.7) alongside lysozyme (pI = 11) from HABP gels with and without Laponite in figure 2. We discuss this data in the text on page 7 as follows:

"Interestingly, and in contrast to its rapid release from HABP alone, HABP+Laponite gels also restricted release of negatively charged albumin over 7 days. This is consistent with studies that describe patch binding onto Laponite surfaces of negatively charged proteins, including BSA, via adsorption of local acidic residues exposed on the protein surface [19]."

4. The authors point to the challenges of "promoting tissue ingress into spaces occupied by the drug-releasing hydrogel". Did the authors assess the degradation rate of the biomaterial?

Please see comments on hydrogel degradation in response to reviewer #1 above.

5. Please discuss any limitations of utilizing laponite in clinical contexts.

Currently, the main limitation for using Laponite in a clinical context is that Laponite is not yet FDA or EMA approved for use as an implantable material. Work is currently in progress in our group, here in Southampton, to undertake the required preclinical safety studies and define the manufacturing requirements ahead of phase 1 first in man safety trials. There are excellent existing reviews on the clinical possibilities of nanoclays **which we now cite** in the introduction in lieu of a formal discussion of this topic which is somewhat beyond the focus of this paper.

Reviewer #3:

1. The loading procedure of the lysozyme into the hydrogels prior to in vitro drug release studies is not clear. There is a need for detailed information.

We thank the reviewer for raising this query. **We have now added further information to the methods section.** We were also concerned that the order of mixing components may alter the release profiles of the composite gels and so undertook a short-term release study to address this question. No effect of loading procedure was observed. **This data is now included as supplementary figure 8 and referenced in the methods section (page 7).**

2. I suggest that more studies should be performed such as SEM analysis so as to study the hydrogel microstructure, FTIR, cytocompatibility studies, calcium deposition on the hydrogel etc.

We have now included additional SEM analysis of the microstructure (Supplementary figure 4) of HA-BP•Laponite physical gels in comparison with chemically cross-linked HABP gels and HA chemically gelled with thiol functionalised Laponite. We agree that this addition serves to strengthen the manuscript.

3. Were the hydrogels allowed to dry after gelation in the syringe mould? What is the nature of the hydrogels?

The syringe molds were sealed to exclude water evaporation so that the volume of water was the same as it was in the preparation, e.g. when aqueous solutions of polymer and Laponite were combined. **We have now made this explicit in the methods section.**

4. The authors need to compare their results with the results reported by researchers that designed similar hydrogels for bone regeneration.

We now include an additional figure setting this result in the broader context of BMP2 dose reduction strategies (Supplementary figure 7). See also our response to reviewer #2 above [1].

5. Did the hydrogel average strength (MPa) fall within the range suitable for bone regeneration?

There is no single predefined range of appropriate mechanical properties for bone regeneration, which depends on the clinical indication and biomechanical strategy employed. Our first target indication for this material is lumbar spinal fusion. The material properties are advantageous in this case as they allow injectable delivery directly to the site (or perfusion into an implant or bone graft material) before setting into a stiff gel that provides a degradable scaffold for subsequent boney ingrowth.

6. I suggest that the title should be revised.

It is not clear why (the rationale implied) or how to revise the title based on this comment. We are pleased with the title and would hope to be able to retain the title at the editor's discretion.

Reviewers' Comments:

Reviewer #1:

Remarks to the Author:

Authors have addressed all the concerns raised by reviewers, so it is recommended for acceptance.

Reviewer #2:

Remarks to the Author:

The authors have adequately addressed my previous comments

Reviewer #3:

Remarks to the Author:

Dear Editor,

The research reported in the manuscript is novel. The conclusions are original. The authors have addressed the comments and suggestions raised by the reviewers. I recommend that the manuscript be accepted for publication.